# Chronic Disruption of the Late Cholesterol Synthesis Leads to Female-Prevalent Liver Cancer

**DOI:** 10.3390/cancers12113302

**Published:** 2020-11-09

**Authors:** Kaja Blagotinšek Cokan, Žiga Urlep, Gregor Lorbek, Madlen Matz-Soja, Cene Skubic, Martina Perše, Jera Jeruc, Peter Juvan, Tadeja Režen, Damjana Rozman

**Affiliations:** 1Centre for Functional Genomics and Bio-Chips, Institute of Biochemistry, Faculty of Medicine, University of Ljubljana, SI-1000 Ljubljana, Slovenia; kaja.blagotinsek@mf.uni-lj.si (K.B.C.); zigaurlep@gmail.com (Ž.U.); greg.lorbek@gmail.com (G.L.); cene.skubic@mf.uni-lj.si (C.S.); peter.juvan@mf.uni-lj.si (P.J.); tadeja.rezen@mf.uni-lj.si (T.R.); 2Rudol-Schönheimer-Institute of Biochemistry, Divison of General Biochemistry, Faculty of Medicine, University of Leipzig, 04103 Leipzig, Germany; madlen.matz@medizin.uni-leipzig.de; 3Medical Experimental Centre, Institute of Pathology, Faculty of Medicine, University of Ljubljana, SI-1000 Ljubljana, Slovenia; martina.perse@mf.uni-lj.si; 4Institute of Pathology, Faculty of Medicine, University of Ljubljana, SI-1000 Ljubljana, Slovenia; jera.jeruc@mf.uni-lj.si

**Keywords:** cholesterol biosynthesis, hepatocellular carcinoma, lanosterol 14α-demethylase (CYP51), sex dimorphism

## Abstract

**Simple Summary:**

Hepatocellular carcinoma is a disease with a variety of molecular triggers and is usually reported to prevail in males. However, after the menopause, the disease is also increasing in the female population. Herein, we discovered that chronic depletion of cholesterol synthesis due to the knock-out of the gene *Cyp51* from this pathway leads to female prevalent hepatocarcinogenesis in aging mice. There is a high similarity between our mouse model and the situation in humans. Multiple deregulated pathways of hepatocarcinogenesis are shared. A female-dependent metabolic reprogramming leading to this type of liver cancer is exposed for the first time and reflects on deregulated cholesterol synthesis as the metabolic trigger. These data are of crucial importance. Despite the higher overall prevalence of hepatocellular carcinoma in males, we need tools and biomarkers to further stratify patients and offer better diagnosis and treatment options to both sexes.

**Abstract:**

While the role of cholesterol in liver carcinogenesis remains controversial, hepatocellular carcinoma generally prevails in males. Herein, we uncover pathways of female-prevalent progression to hepatocellular carcinoma due to chronic repression of cholesterogenic lanosterol 14α-demethylase (CYP51) in hepatocytes. Tumors develop in knock-out mice after year one, with 2:1 prevalence in females. Metabolic and transcription factor networks were deduced from the liver transcriptome data, combined by sterol metabolite and blood parameter analyses, and interpreted with relevance to humans. Female knock-outs show increased plasma cholesterol and HDL, dampened lipid-related transcription factors FXR, LXRα:RXRα, and importantly, crosstalk between reduced LXRα and activated TGF-β signalling, indicating a higher susceptibility to HCC in aging females. PI3K/Akt signalling and ECM-receptor interaction are common pathways that are disturbed by sex-specific altered genes. Additionally, transcription factors (SOX9)2 and PPARα were recognized as important for female hepatocarcinogenesis, while overexpressed *Cd36*, a target of nuclear receptor RORC, is a new male-related regulator of ECM-receptor signalling in hepatocarcinogenesis. In conclusion, we uncover the sex-dependent metabolic reprogramming of cholesterol-related pathways that predispose for hepatocarcinogenesis in aging females. This is important in light of increased incidence of liver cancers in post-menopausal women.

## 1. Introduction

Metabolic abnormalities result in metabolic associated fatty liver disease (MAFLD), previously termed (NAFLD), which affects over 30% of the western population and has no approved drug therapies [1]. MAFLD can further develop into hepatocellular carcinoma (HCC) and both diseases are predicted to increase in the following decades [2], not only in men but also in women [3].

Among the lipid factors associated with progressive liver damage is also cholesterol. The cellular cholesterol concentrations are fine-tuned by a balance of synthesis, uptake, and efflux [4]. Genes of cholesterol synthesis differ in their susceptibility to loss of function mutations and can affect cell metabolism in different manners, from promoting cell survival towards manifesting in malignancies [5,6]. Linking cholesterol synthesis and liver disease remains controversial. Cholesterol synthesis was shown to support the growth of HCC lesions after depletion of fatty acid synthesis [7,8], indicating crosstalk between fatty acid and cholesterol pathways in carcinogenesis. Overexpression of the squalene epoxidase enzyme of cholesterol synthesis was identified as a driver of NAFLD-induced HCC [9]. On the other hand, blocking cholesterol synthesis at the lanosterol 14α-demethylase (CYP51) step leads to the prepubertal onset of liver injury with ductular reaction and fibrosis and a pronounced male prevalent liver dysfunction before puberty [10]. In young adults, the liver damage was more prominent among female mice due to diminished cholesterol esters and elevated sterol substrates [11].

The sex disparity in liver pathologies is far from conclusive. While some studies do not describe sex as a cofounding factor [7,9,12,13], there is increasing evidence that after the menopause MAFLD occurs at a higher rate in women compared to men [14]. Female and male livers are metabolically distinct organs, which was demonstrated also in mathematical models of the liver with unique regulators of sex-specific metabolic outcomes [15]. A recent review concluded that clinical and epidemiological studies frequently fail to address sex differences appropriately [14].

This study aimed to address the long-term effect of disrupted liver cholesterol synthesis where the *Cyp51* knock-out (KO) model of aging mice was applied in light of the relevance to humans. Surprisingly, the tumors developed spontaneously mainly in female aging mice. We identified novel metabolic pathways and transcription factors that can be linked to metabolism associated with female-prone hepatocarcinogenesis. We provide mechanistic insights into the metabolic reprogramming in this type of liver cancer, with special attention to that of females.

## 2. Results

### 2.1. Long-Term Hepatocyte Deletion of Cyp51 from Cholesterol Synthesis Results in HCC

We wanted to understand how the long-term depletion of cholesterol synthesis influences liver physiology. We expected that due to the damaged livers in young adulthood [16], the mice would have a shortened life span. In contrary to expectations, mice lived at least until 24 months (M). Their livers did not become cirrhotic but started to develop liver tumors between 12M and 24M of age. Hepatomegaly was more pronounced in females’ KOs (Appendix A). The histopathological examination of livers identified moderate to severe ductular reaction bridging the adjacent portal tracts, accompanied by mild inflammation (predominantly mononuclear cells; arrows in Figure 1A), and fibrosis. Importantly, fibrosis was more pronounced in females (Sirius red staining (SR); Figure 1B). Yellow pigment likely representing lipofuscin (arrow in Appendix A) was observed in areas of ductular reaction, sometimes surrounded by other hepatic cells, evaluated as macrophages (stars in Figure 1A).

The incidence of tumors by sex end genotype is presented in Appendix A. Liver tumors were observed initially in 12M female and 18M male KO mice (Figure 2Aa). They were defined as macroscopically visible nodules (>1 mm) of various colours. At 24M, the incidence of tumors was 77.8% for females and 50% for males and were observed only in KO mice. In wild-types (WT), a single tumor was found in a 2-year male, histologically described as adenoma (Appendix A). At 24M, the KO mice showed a decreased body weight compared to controls, significantly higher liver weight, and elevated relative liver weights. This was especially true for female mice with liver tumors (Appendix A and Appendix A). Tumors were histologically classified as eosinophilic/clear cell nodules, hepatocellular adenomas, or HCCs according to published criteria [17]. None of the tumors showed histological features of biliary tumor, e.g., biliary adenoma or cholangiocarcinoma.

Histologic evaluation is often sufficient for distinguishing the two types of primary hepatocellular tumors (HCC and cholangiocarcinoma), except in the case of poorly differentiated tumors. For further molecular analyses, we selected only tumors histologically classified as HCC, with features such as the absence of portal tracts and hepatic lobules, thickened hepatic plates, solid growth, focal tubular or pseudoglandular structures, focal necrosis, numerous apoptotic cells, and presence of cellular and nuclear polymorphism with brisk mitotic activity (Figure 2Ab,c).

Histologic evaluation is the gold standard in distinguishing the two most common primary hepatocellular tumors, namely hepatocellular carcinoma (HCC) and cholangiocarcinoma, except in the case of poorly differentiated tumors. Immunohistochemistry is a useful tool in determining the histogenesis of tumours. However, no single marker has been found that is completely specific for HCC. Among many different HCC markers, we decided for glutamine synthetase (GS), as it was recognized as the single best immunostain for identifying hepatocellular tumors in mice in a recent study [18]. As expected for normal liver tissue, positive GS staining was observed in pericentral hepatocytes (PC), but not in midzonal (M) or periportal (PP) hepatocytes (Figure 2Ba). Different staining patterns were observed in tumor tissue of *Cyp51* KO mice, from scattered focal (Figure 2Bb), to diffuse staining of neoplastic hepatocytes (Figure 2Bc).

### 2.2. Sex-Dependent Progression from Metabolism Associated Liver Disease towards HCC

To learn about molecular changes based on long-term metabolic insult—the ablation of *Cyp51* in hepatocytes—we assessed the hepatic expression of particular genes linked to cholesterol homeostasis during aging and measured multiple metabolites and markers in the blood of females and males. The results are presented graphically in Figure 3, with the most statistically significant changes between the groups (male, female, WT, *Cyp51* KO) over all time points (12M, 18M, 24M) in Appendix A. At the expression level, deregulation of cholesterol synthesis genes (Appendix A) was, during aging, more pronounced in female livers (Figure 3A: *Cyp51* KO—violet line, WT—black line) compared to those of males (Figure 3A: *Cyp51* KO—green line, WT—blue line). The key regulatory gene of cholesterol synthesis *Hmgcr* is, over time, overexpressed in the KOs, particularly in females at 12M. While at 12M *Cyp51* is downregulated in the livers of *Cyp51* KO, this is not the case anymore at older age. We were initially intrigued by this finding. However, the *Cyp51* is knocked-out only in the hepatocytes and not in other cell types of the liver that seem to complement for the *Cyp51* loss and increase its expression. If we look into the cholesterol synthesis gene expression in the tumors at 24M (Appendix A), all genes except *Cyp51* were upregulated, as expected for feedback regulation after the block in cholesterol synthesis, again significantly only in the female knock-outs. Investigation of bile acid (BA) synthesis pathway (Figure 3B, Appendix A) showed the expression of *Cyp8b1*, *Cyp7a1* and *Cyp27a1* decreased in the KOs, with no significant sex differences. At 24M in tumors, *Cyp7a1* is significantly downregulated in the female KOs (Appendix A), indicating a drop in the classical pathway of BA synthesis.

A major sex disparity was observed when monitoring the blood plasma biochemical parameters during aging. Total plasma cholesterol, free fatty acids (FFA), HDL (high-density lipoprotein cholesterol), and triglycerides (TG) showed opposite trends between sexes. Cholesterol and HDL were higher in KO females and were increasing by age, while they were lower in KO males compared to male controls, and decreasing with age. A significant increase in ALT in female *Cyp51* KO mice is an additional indication of chronic hepatic injury (Figure 3C and Appendix A), which is more prominent in females.

To quantify the primary molecular insults in the livers, we measured sterol intermediates of cholesterol synthesis together with cholesterol (Figure 3D and Appendix A). In the KO mice with inactivated CYP51A1 enzyme, its substrates lanosterol and dihydrolanosterol were highly elevated (~100-fold and ~500-fold, respectively), proving disrupted cholesterol synthesis. Dihydrolanosterol was higher than lanosterol, indicating that 24-dehydrocholesterol reductase (DHCR24) enzyme functions normally. Downstream the pathway, desmosterol was not significantly different between *Cyp51* KO and WT mice; however, zymostenol was higher in *Cyp51* KO livers. Small or no changes between the sexes were observed in intermediate sterols. In contrast to that, sex represents a statistically significant parameter for liver cholesterol in the interaction analysis between sex and age. Similarly as for plasma cholesterol, hepatic cholesterol is also increased in female mice during ageing, while this was not the case for males (Figure 3C,D and Appendix A).

### 2.3. Transcriptome Analysis Identified Sex Differences in Gene Expression Linked to Hepatocarcinogenesis

We used Affymetrix microarrays for gene expression profiling of mouse liver samples from 24M *Cyp51* KO mice (24M KO) and controls (24M WT). We performed an analysis of liver tumors and the surrounding tissue that is responsible for tumorigenesis. To monitor the progression of tumorigenesis, data was compared to precancerous stages at 19 weeks of age (19W KO *versus* 19W WT) [16] and discussed in view of the early processes towards carcinogenesis (Appendix A).

Hierarchical clustering of differentially expressed (DE) genes showed a clear difference between 24M WT and 24M KO mice for both sexes (Figure 4A). Importantly, a clear sex imbalance in the number of DE genes was observed (Figure 4B), with more genes (63.4%) deregulated in the female livers, including *Ctnnb1*, *Gsk3b*, *Nova1*, *Lmna*, *Gstp1*, and *Mup20* (Appendix A). Among the DE genes deregulated only in males (11.6%) were *Trp53, Ezh2*, *Foxa1*, and *Rb1*. In both sexes, a majority of DE genes were positively enriched (56% in females and 68% in males) (Figure 4B). Using NetworkAnalyst [19], the functionally enriched networks of DE genes showed that positively enriched genes were mostly involved in pathways related to cancer and the extracellular matrix (ECM)-receptor interaction pathway, while genes from metabolic pathways were negatively enriched, especially among female KOs (Figure 4C).

Additionally, we selected the top 10 statistically sex-specific significant down- and up-regulated genes (Appendix A), whose biological functions were examined. Lipid metabolism genes are among the most down-regulated, however, with some distinct differences in lipid related pathways in 24M KO females and KO 24M males. Most down-regulated genes in KO males are from metabolism of fatty acids, such as fatty acid elongation, and PPAR signalling. In the female KOs, we observed downregulation of steroid hormone biosynthesis and related metabolic pathways, bile secretion, and also fatty acid metabolism (arachidonic acids). Drug metabolism is also downregulated in KO females. Retinol and linoleic metabolism were among top 10 in both sexes.

Oppositely, top 10 up-regulated genes in male KO liver are mainly involved indifferent cancer pathways, PPAR signalling and ECM receptor interactions, while the female list includes other upregulated genes in PPAR signalling, Wnt signalling and diabetes pathways.

### 2.4. Enriched KEGG Pathways and Transcription Factors (TFs) as Triggers of Female Prevalent Hepatocarcinogenesis

From 323 KEGG pathways, we discovered 209 that were enriched in livers of precancerous 19W KO mice and 230 in 24M KO mice with tumors, compared to 24M WTs (Table 1 and Appendix A). The enriched pathways include the activation of HCC, apoptosis, PI3K/Akt signalling, pathways in cancer, and proteoglycans in cancer. The ECM-receptor interaction pathway was observed to be the most positively enriched in 19W KO and 24M KO females and males (Table 1), which is in line with the histological findings. Chronic injury induces numerous molecular alterations in hepatocytes and causes a gradual accumulation of extracellular matrix components (ECM) that might lead to a complete architectural reconstruction of hepatocytes and tumor development.

We discovered multiple signalling pathways that describe the sex-dependent metabolic reprogramming in liver cancer. Among them, TGF-β signalling was significantly elevated only in *Cyp51* KO female livers, both at the precancerous stage and also later (Table 1). Additionally, the Wnt signalling and circadian entrainment pathways were activated only in the cancers of KO females (Table 1). We further explored whether these pathways could serve as triggers for female hepatocarcinogenesis in the *Cyp51* KO model. The selected protein markers of the TGF-β and Wnt signalling were evaluated by qPCR and IHC (Appendix A). The TGF-β showed a positive reaction, particularly in female *Cyp51* KO mice, indicating this is a good biomarker to follow progression towards HCC, particularly in females.

Among lipid metabolic pathways, we found fatty acid degradation and primary bile acid synthesis negatively enriched in KO females at 19W and 24M, while this was the case only for 24M KO males (Table 1), suggesting that the metabolic reprogramming of fatty acid and BA metabolism is active earlier in female carcinogenesis.

Considering the general alteration of DE genes and KEGG pathways in hepatocarcinogenesis, we also investigated the transcriptional landscape by transcription factor (TF) enrichment analysis. This analysis indicated 321 enriched TFs in the precancerous 19W KO livers and 142 in tumor containing 24M KO livers, both in comparison to WT littermates (Appendix A). The enriched TFs with specific functions are listed in Table 2. From the positively enriched transcriptional activity SP1, C-JUN, JUND, C-FOS, AP1 and C/EBPβ were enriched already in the precancerous stage in both sexes, while PPARα, (SOX9)2 and NFATC1 are positively enriched only in females, starting already at the precancerous stage. HIF1α was also enriched in females in the precancerous stage, and together with SOX9, also in tumors of both sexes. The list of negatively enriched TFs is shorter. Downregulated CLOCK is characteristic for HCC, irrespective of the sex. HNF4α and RORC are negatively enriched in females already at the precancerous stage and in both sexes in tumors. NR1B1 is the only TF downregulated in the precancerous male livers and later in tumors of both sexes. LXRα:RXRα is a female-specific HCC marker.

When we considered the top-10 sex-specific enriched TFs based on the fold-change (Appendix A), additional information was gathered. PPARα and the oncogenic isoform (SOX9)2, which were exposed previously (Table 2), are among top 10 positively enriched TFs by fold-change in females. FXR is, by fold-change, the most downregulated female-specific TF in HCC that was not exposed in the overall TF deregulation of Table 2. LXRα:RXRα were, again, among the most negatively enriched TFs in female tumors (Appendix A). SOX9 is a part of the Wnt signalling and is expressed in the progenitor/precursor cells during hepatocyte regeneration following liver injury [20]. Expression of heterodimer complex LXRα:RXRα correlates with the amount of the retinoic acid and cholesterol derivates [21], which is line with diminished activity in KO female mouse livers. It was shown that chronic administration of PPARα agonists leads to liver tumor development in mice [22]. However, it is still unclear how this receptor changes in the human liver cancer tissue. An investigation of PPARα mRNA expression in human HCC indicated an increased mRNA expression of PPARα and its target genes. These results point out the possible association of activated PPARα with human carcinogenesis [23]. NFATC1, which was not on the list of sex-specific top 10 TFs, was also indicated as female-specific TF (Table 2). The NFATC1 with calcineurin overexpression plays an important role in the proliferation of HCC [24]. NFATC1 transcription can be activated by AP1 (including FOS–JUN), the activation of which has been already described in T cells [25].

RORC (RORγ) was one of the TFs in our focus since sterols after lanosterol are ligands of this transcription factor [26]. Downregulation of RORC signalling is thus important due to lowered intermediate sterols upon the *Cyp51* block.

To validate whether downregulation of RORC has effects in hepatocarcinogenesis, we evaluated the expression of downstream genes regulated by this transcription factor. We found a majority of RORC target genes decreased in *Cyp51* KO mice (Appendix A), since RORC serves as their activator. However, the RORC target *Cd36* was upregulated since RORC is known to repress this gene [10]. The RORC target *G6pc*, which is involved in the regulation of the PI3K/Akt signalling pathway, was negatively enriched in female KO mice, representing a part of the female metabolic reprogramming (Appendix A).

A summary of sex-dependent reprogramming of metabolic pathways and corresponding transcription factors is shown in Figure 5. Briefly, Figure 5 indicates connections between signalling pathways, TFs and cellular processes, which were deregulated in both sexes (top half of figure) and specific for female KOs (bottom half of figure), leading to the inhibition of basal metabolism and an increased possibility for tumor development.

The TF networks were visualized with STRING [27] to determine their interactions. Different colours of nodes represent biological processes in which enriched TFs of *Cyp51* KO mice are involved. Surprisingly, SP1 and JUND deregulated in precancerous and tumor livers of both sexes represent nodules with most protein–protein associations linked to hepatocarcinogenesis (Appendix A).

### 2.5. Metabolic and Transcriptional Changes after Disrupted Cyp51 from Cholesterol Synthesis Align with Hepatocarcinogenesis in Humans

We performed a comparative analysis of sex-dependent metabolic adaptations upon inactivation of CYP51A1 in the hepatocytes in light of the human literature data. Figure 6 represents the proposed carcinogenesis mechanism in our mouse model with exposed signalling, their specific sex-dependent targets and possible transcription regulators that might offer clues for expanding the understanding of the human sex-dependent HCC carcinogenesis. In the mouse model, the chronic metabolic insult of disrupted cholesterol synthesis (accumulation of hepatic sterols lanosterol and dihydrolanosterol) resulted in a variety of sex-specific metabolic adaptations. The hepatic cholesterol was increased in KO females and decreased in KO males. This observation serves as one of the important features in cancer cells. As fast proliferating cells, cancer cells require high levels of cholesterol for membrane biogenesis and other functional needs. For example, the cholesterol-derived oncometabolite 6-oxo-cholestan-3β,5α-diol was enriched in patients with breast cancer and subsequently promotes tumor growth [6].

As shown in Figure 6, disturbance of cholesterol synthesis leads to the repression of selected nuclear receptors (FXR, LXRα:RXR, LRH1 and NR1B1:RXRα) in female KO hepatocytes, which are increasingly appreciated targets in the prevention of inflammation and fibrosis [6,28]. The positive enrichment of PPARα, observed only in females, was also confirmed in humans with NAFLD [21], but its role in HCC patients remains unclear [29]. FXR negative enrichment and downregulation is also in line with the human HCC progression. Negative enrichment of FXR in females and downregulation of its target gene *Ndrg2* is in line with the human data in patients and confirmed tumor suppressor role of FXR and NDRG2 in human cell lines [30].

Aligned with the latest human literature, metabolic and transcriptional changes of disturbed cholesterol metabolism in females pinpoints on a new sex-dependent mechanism, which has not been revealed so far. We proposed that a higher incidence of HCC in females results from the crosstalk of downregulated transcriptional activity of LXRα and activation of TGFβ signalling. Positive expression of TGF-β signalling targets *Gstp1*, *Tgfrb1*, *Tgfrb2* and *Bmp7* was identified only in female KO mice. Among several human malignancies, TGF-β in humans regulates the progression of female HCC through the promotion of PI3K/Akt signalling [31] and ECM-receptor interaction [32]. PI3K/Akt signalling and ECM-receptor interaction are common pathways that are disturbed by altered gene expressions in human cancers [33,34]. In our mouse model, both pathways were activated, but through different positively enriched genes in females (*Col5a2* and *Mmp9*) compared to males (*Cd36*). In the context of lipid homeostasis, Raichur S et al. showed the opposite expression patterns, where repressed RORC activity induces expression of *Cd36.* In both animal and human studies, increased *Cd36* expression in metabolically active tissue has been implicated in the regulation of fatty acid metabolism and lipid accumulation [35,36]. In HCC patients, the number of fatty acid responsive TFs, including HIF1α and FOS, can transcriptionally upregulate *Cd36*, which is closely associated with induction of epithelial-mesenchymal transition (EMT) [37].

Moreover, *Gck,* which is known as a one of RORC targets, was downregulated in Rorc^−/−^ mice [38], while in female *Cyp51* KO livers, it was upregulated. The identity of all TFs regulating *Gck* promoter activity has not been fully elucidated. One of the possible co-regulations of the liver *Gck* expression may be through the insulin, which can increase *Gck* mRNA via PI3K/Akt signalling, regulating the HIF1α [39] that were both found to be activated in KO mice.

In humans, the expression of SOX9 correlates with the severity of liver fibrosis patients [40] and correlates with the progression to cirrhosis and HCC [41]. In our case, the activated TF (SOX9)2, and its targets *Ctgf* and *Nfe2l1* (Appendix A), were deregulated only in female KOs.

## 3. Discussion

The hallmark of cancer cells is metabolic reprogramming, causing new cellular demands in selective survival and growth [42]. Cholesterol is a precursor of steroid hormones and bile acids and an essential component of cellular membranes. Its circulating levels are under healthy homeostatic conditions regulated by a balance between cellular cholesterol synthesis, dietary intake, and removal of excess cholesterol [43].

*CYP51A1* is a gene from the late (post-squalene) part of the pathway [44]. The disruption of this gene in hepatocytes starts with morphological alterations, such as ductular reaction accompanied by mild inflammation. We show herein that this can end with malignant liver tumors in ageing mice that are first observed at 12M in females and 18M in males, with the highest incidence in the 24M female *Cyp51* KO mice. In addition, humans show sex-dependent differences in liver pathologies. Women more commonly present with acute liver failure, benign liver lesions, autoimmune hepatitis, and toxin-mediated hepatotoxicity. Oppositely, malignant liver tumors and viral hepatitis are less commonly observed in women [45]. However, non-alcoholic steatohepatitis (NASH), as a more serious form of MALD, more commonly progresses to HCC in women [46]. Generally, women are less prone to developing HCC than men (due to hormone-dependent resistance), while after the menopause a sharp increase in HCC incidence is witnessed [47].

Based on our data, we propose that the repression of CYP51A1 enzyme can contribute to HCC development. In humans, whole-body deletion of *CYP51A1* is not reported, since this gene is essential and is embryonically lethal in the mouse [48]. Only two studies are reporting the *CYP51A1* deleterious mutations, which resulted in infant liver failure and death [49,50]. Numerous polymorphisms in *CYP51A1* have been associated with different phenotypes in humans [51], but the consequences of *CYP51A1* heterozygosity in human hepatocarcinogenesis need yet to be established. *CYP51A1* has a low mutation rate, lower than other cholesterol synthesis genes [52]. The *CYP51A1* rs37417517 variant was found to be associated with the female-specific breast cancer and variant rs229188 with female genetics [53,54]. Additionally, the COSMIC database revealed the presence of somatic *CYP51A1* point mutations in tumor samples, some also with a predicted deleterious effect on activity (Appendix A).

The synthesis of cholesterol is a housekeeping pathway, which includes more than 100 genes associated with cholesterol synthesis and its regulation [55]. The block in cholesterogenic *Cyp51* in KO mice is in our case specific in hepatocytes. Thus, about 40% of *Cyp51* gDNA remained unchanged in the livers of KO mice, which lead to an approximally 60% decrease in mRNA and an 80% decrease in the CYP51 protein. Lorbek et al. indicated the small foci of stained periportal hepatocytes that potentially originated from the oval cell compartment by immunohistochemical staining of CYP51. Thus, the high intensity of ductular proliferation in ageing mice may explain an increase in *Cyp51* mRNA in female and male KO livers from 18M onward [16]. Oppositely, the drop in expression of other genes of cholesterol synthesis could be a result of hepatic inadequacy due to chronic liver disease, which was also shown in fibrotic patients [56].

The latest studies indicate that post-squalene intermediates of cholesterol synthesis are also signalling molecules. Inhibiting different enzymes from the pathway results in the accumulation of sterol intermediates with different cellular effects. Sterol products can activate RORC signalling [26], promote oligodendrocyte formation [57], or activate LXRα and inhibit EGFR signalling [58]. Inhibitors of lanosterol synthase were identified as potential anticancer drug targets [59] while the accumulation of lanosterol can cause degradation of 3-hydroxy-3-methyl-glutaryl-coenzyme A reductase (HMGCR) [60]. Our sterol measurements show significant accumulation of CYP51A1 substrates lanosterol and dihydrolanosterol, with no sex differences. We observed lower amounts of desmosterol and cholesterol in KO livers. The presence of desmosterol in the liver may be explained by compensatory effects on lower levels of DHCR24, as it was shown in the brain [61]. The liver cholesterol may be obtained from the diet, to compensate the defected de novo synthesis of cholesterol [62]. The increase in cholesterol in female KO livers may arise due to the cholesterol uptake from plasma, since cholesterol is required for tumor cell proliferation and growth [56,63].

The main difference between the genotypes is likely in the accumulation of sterol substrates rather than a lack of downstream sterols, while the difference between the sexes is shown by the sex-dependent transcriptome response to sterol imbalance. Due to the lack of sterols between lanosterol and zymosterol, which are proposed RORC natural ligands, we observe at 24M the repression of the RORC signalling pathway in both sexes. The RORC-dependent mechanisms in cancerogenesis were recently described. Oh et al. reported that RORC expression is decreased in basal-like subtype cancers and inversely correlated with histological grade and cancer drivers in breast cancer cohorts [64]. In breast cancer cell lines, RORC was shown to repress TGF-β/EMT-signalling and cancerogenic pathways, which indicates that, also in *Cyp51* KO mice, decreased activity of RORC could contribute to increased TGF-β/EMT signalling. There is no data yet that would link SOX9 expression to sterols. (SOX9)2 is upregulated, particularly in KO females, and further research is needed to evaluate whether this is a direct effect of accumulating lanosterol and dihydrolanosterol.

Injured hepatocytes in *Cyp51* KO mice produce different ECMs, exemplified by the enrichment of laminin and collagen genes (*Col4a1*, *Lamc1*, etc.) and a strongly positively enriched ECM-receptor interaction pathway in hepatocarcinogenesis of both sexes. In some way, these alterations could contribute to microenvironmental reorganization and induction of EMT that positively influences tumor progression [32]. A higher rate of cancer progression in female *Cyp51* KO mice could be explained by the female-specific interplay of positively enriched TGF-β signalling. Activated TGF-β represents a central regulator of chronic liver disease and could contribute to the progression of all disease stages, both in animal models and in humans [65]. Interestingly, important crosstalk between TGF-β and *Cyp51* gene expression has been shown in the Tgf-βfl/fl;Wnt-Cre mouse model [66].

Furthermore, the results of our analysis identified the key transcriptional regulators above the differentially expressed molecular pathways: positively enriched SOX9, SP1, and negatively enriched RORC. SOX9 is an important candidate for the cancer marker required for activation of TGF-β/Smad signalling [67]. SOX9 expression, especially its isoform 2 ((SOX9)2), can predict liver disease progression [40], but was so far not described as being associated more with the female type of carcinogenesis. Similarly to SOX9, the tumor progression driver SP1 is also a key factor for the induction of TGF-β during inflammation [68] and its overexpression in *Cyp51* knock-outs relates to EMT. The impact of higher disease progression in females is reflected also in the interplay of transcription regulator SMAD3 on TGF-β signalling [69].

Increased plasma total cholesterol and HDL, and decreased plasma TG and FFA, are observed only in female KO mice. This is a possible effect of induced sex hormone-binding globulin (SHBG) by specific targets of an estrogen signalling pathway [70]. SHBG positively correlates with HDL and negatively with TGs and has been already associated with different cancers in ageing humans [71,72]. Supporting other hepatocarcinogenesis studies [73,74], the outcome of altered cholesterol and BA homeostasis was a chronic liver injury in our *Cyp51* KO mouse model with widespread fibrosis, inflammation, and malignant transformation due to toxic sterol accumulation. Additionally, sterol accumulation, in our case the consequence of inhibition of CYP51A1 [16], could be a cause of the observed widespread increase in hepatic progenitors (also known as a ductular reaction) that may also differentiate into mature hepatocyte-like cells with oncogenic potential [75].

## 4. Materials and Methods

Detailed materials and methods are described in Appendix B. The experimental design of the study is shown in Figure A1.

### 4.1. Animal Study and Samples Collection

The generation of homozygous hepatocyte-specific *Cyp51* knock-out (*Cyp51* KO) mice has been reported previously [10,16]. Animals were bred under a 12 h light and dark cycle (7:00 a.m. until 7:00 p.m. light), at 22 ± 1 °C, humidity 55 ± 10%, on standard rodent chow (diet 1324, Altromin, Lage, Germany) and acidified tap water (pH = 3) *ad libitum*. They were housed in groups of 3–5 mice per cage. The results of routine health microbiological monitoring are shown in Table A1. Mice were sacrificed at different ages (12M, 18M and 24M) with cervical dislocation between 11 and 13 h (to minimize the circadian influence), after a 4–6 h of fasting (food withdrawal at 7 h a.m.). Blood was taken immediately after cervical dislocation from the right ventricle, collected into heparin-coated Vacuette MiniCollect^®^ 1 mL Plasma Tubes (Greiner Bioone, Frickenhausen, Germany) and centrifuged for 4 °C, 3000× *g* for 15 min. Mice were weighed, organs (liver, kidney, spleen, heart, gonads) removed and patho-morphologically examined. For histology, the left lateral lobus was fixed in 4% formalin and embedded in paraffin for further histological analysis. Other parts of livers were cut into thin slices and snap-frozen in liquid nitrogen. The material was freshly frozen and stored at −80 °C for subsequent analysis.

### 4.2. Ethics Statement

Mouse experiments were approved by the Veterinary Administration of the Republic of Slovenia (license numbers 34401-31/2011/4 and 34401-52/2012/3) and were conducted in agreement with the National Institute of Health guidelines for work with laboratory animals, national legislation and Directive 2010/63/EU on the protection of animals used for scientific purposes.

### 4.3. Histological Analysis

For standard histological analyses, formalin-fixed paraffin-embedded tissue was used, sectioned to 4–5 µm and deparaffinized at 70 °C for 10 min and washed in two changes of Xylene, 100% ethanol, 95% ethanol, 70% ethanol and twice in acidified water. Hematoxylin and Eosin (H&E) stained liver sections of each mouse were scored for size, shape, and polymorphism of the hepatocytes, granulation of cytoplasm, presence of portal and parenchymal inflammation, ductular reaction, cholestasis, and presence of any type of nodule. The intensity of inflammation and ductular reaction was graded as: 0—normal, +—mild, ++—moderate and +++—severe. The extent of ductular reaction was assigned: p—proliferation around portal tracts, p-mz—proliferation extending from one portal tract to the middle zone, and p-p—bile ducts were extending from one portal field to another (bridging). For evaluation of liver fibrosis, Sirius red (SR) staining was performed by incubating deparaffinized sections in 0.1% Sirius Red solution for 1h (SR; 0.1% direct red 80, 1.2% picric acid in water), briefly destaining in diluted acetic acid, dehydrating in 70% ethanol, 95% ethanol, 100% ethanol and twice in Xylene and fixed with Roti Histokitt II (Carl Roth GmbH + Co. KG, Karlsruhe, Germany). Fibrosis in SR stained liver samples were graded: 0 as absent, + mild periportal, ++ moderate, and +++ as very strong fibrosis with bridging. Tumors, identified as macroscopically visible nodules (>1 mm) of various colors observed at autopsy, were histologically classified based on Mouse tumor classification criteria [17] as eosinophilic or clear cells nodules, adenomas, cholangiomas, cholangiocellular carcinoma or hepatocellular carcinoma (HCC).

### 4.4. Immunohistochemistry

To evaluate the expression of selected markers in liver samples submitted to the microarray experiment, immunolabeling for glutamine synthetase, β-catenin, and TGF-β1 was performed. Liver tissue sections were deparaffinized and subjected to degrading alcohol gradient and treated for antigen retrieval before staining. Afterwards, 3% hydrogen peroxide (H_2_O_2_) was used for endogenous peroxidase quenching, non-specific staining was blocked using 5% normal goat serum (Sigma-Aldrich, St. Louis, MO, USA) for 1 h at room temperature. Incubation with primary, anti-glutamine synthetase (BD Biosciences, San Jose, CA, USA; 1:1000 dilution), anti-β-catenin (BD Biosciences, San Jose, CA, USA; 1:1000 dilution) and anti-TGF-β1 (Promega, Madison, WI, USA; 1:500 dilution) antibodies in 1% goat serum in 0.1% Tris buffered saline with Tween 20 (TBST) was performed overnight at 4 °C. We used the DAKO EnVision Detection System (Agilent Technologies DAKO, Glostrup, Denmark) for antibody detection. The sections were subsequently counterstained with hematoxylin.

### 4.5. Analysis of Plasma Parameters

Total and HDL cholesterol, free fatty acids (FFA), triglycerides (TG), alanine aminotransferase (ALT), and aspartate aminotransferase (AST) were analyzed with the Architect ci8200 analyzer (Abbott Diagnostics, Abbott Park, IL, USA). The concentration of lipid parameters were measured as mmol/L and a linear regression model Age+Genotype+Sex+Age:Sex was used to fit the concentrations of individual lipids. The activity of ALT and AST were given in μkat/L and a linear regression model Age+Genotype+Sex+Genotype:Sex was used to fit their activity individually. ANOVA type III test was used to estimate the statistical significance of the effects and the Holm method was used to control the family-wise error rate (FWER) at α = 0.05.

### 4.6. Liver Sterol Analysis

Sterols isolation: Sterol intermediates were isolated with the protocol described previously [76]. Folch solution with samples corresponding to 80 mg of liver tissue was transfer to fresh tubes and 200 ng of internal standard Lathosterol-D7 (Avanti Polar Lipids, Alabaster, AL, USA) was added. Final samples were dissolved in 250 µL of methanol and transferred to HPLC vials for analysis. LC-MS analysis was performed as described [77]. Briefly, sterols were separated on combined columns of Luna^®^ 3 µm PFP (2) 100 Å (100 mm and 150 mm length) on a Shimadzu Nexera XR HPLC. For mobile phase, 80% methanol, 10% water, 10% 1-propanol, and 0.05% Formic acid were used in the isocratic condition. HPLC was coupled with the Sciex Triple Quad 3500 mass spectrometer for detection. For ionization, APCI (Atmospheric-pressure chemical ionization) was used in positive mode and detection was made in MRM (Multiple reaction monitoring) mode. The sample concentration was normalized on internal standard (Lathosterol-d7) and sterol concentrations calculated to corresponding standards from Avanti Lipids. We measured the following sterol intermediates, with MRM in brackets: Zymosterol* (367/215), 24-dehydrolathosterol* (367/215), 7-dehydrodesmosterol* (365/199), Desmosterol (367/215), Zymostenol (369/215), Lathosterol-d7 (376/215), FF-MAS* (393/214), T-MAS* (395/243), Cholesterol (369/215), Lanosterol (409/191) and 24,25-dihydrolanosterol (411/191). *Sterol concentration was too low to quantify.

### 4.7. Gene Expression Profiling and Network Analysis

Transcriptome analysis of liver samples of *Cyp51* WT and KO mice of both sexes was performed using Affymetrix GeneChip™ Mouse Gene 2.0 ST arrays and validated by qPCR (primers listed in Appendix A; further details in Appendix B. The same mice and groups were used for plasma analysis (Table A2). For qPCR, a linear regression model Age+Genotype+Sex+Age:Genotype was used to fit normalized expression data of cholesterol-related genes *Hmgcr*, *Sqle*, *Lss*, *Cyp51*, *Nsdhl* and *Dhcr7*, and a reduced additive model Age+Genotype+Sex was used to fit data of fatty acid-related genes *Cyp7a1*, *Cyp8b1* and *Cyp27a1*. ANOVA type III test was used to estimate the statistical significance of the effects and Holm method was used to control FWER at α = 0.05.

A total of 20 Affymetrix GeneChip™ Mouse Gene 2.0 ST Arrays (Affymetrix, Santa Clara, CA, USA) with liver samples were processed. From each of the *Cyp51* KO livers, two samples were taken: tumor and surrounding tissue. Altogether, 6 tumor and 6 corresponding surrounding-tissue samples were obtained from *Cyp51* KO mice (Table A3). The data were analysed using R/limma, controlling the false discovery rate at α = 0.05, and deposited to Gene Expression Omnibus (GEO) under accession number GSE127772. KEGG PATHWAY and TRANSFAC databases were used for functional enrichment studies. Gene expression data from 19 weeks (W) [16] WT and KO mice of both sexes were used to determine processes that contribute to tumor progression from 19W towards 24M. Network diagrams were created using NetworkAnalyst and STRING. The proposed mechanistic scheme was designed from data obtained from mice experiments and aligned to human literature data to increase their relevance.

## 5. Conclusions

The hallmark of cancer cells is metabolic reprogramming, causing new cellular demands in selective survival and growth. Metabolic reprogramming of hepatic cholesterol synthesis leads to female prevalent hepatocellular carcinoma during aging. Inactivation of *Cyp51* affected multiple signalling pathways and transcription factors in a sex-dependent manner, resulting in metabolic-related HCC. The female-specific cholesterol-related mechanism of hepatocarcinogenesis is very similar to the human data and is in line with the fact that the liver is a sex-dimorphic metabolic organ and a major organ of cholesterol homeostasis. Since hepatocellular carcinoma is increasing in post-menopausal women, the female-specific aspects of liver cancer progression, diagnosis and treatment should become a central focus of research.

## Figures and Tables

**Figure 1 cancers-12-03302-f001:**
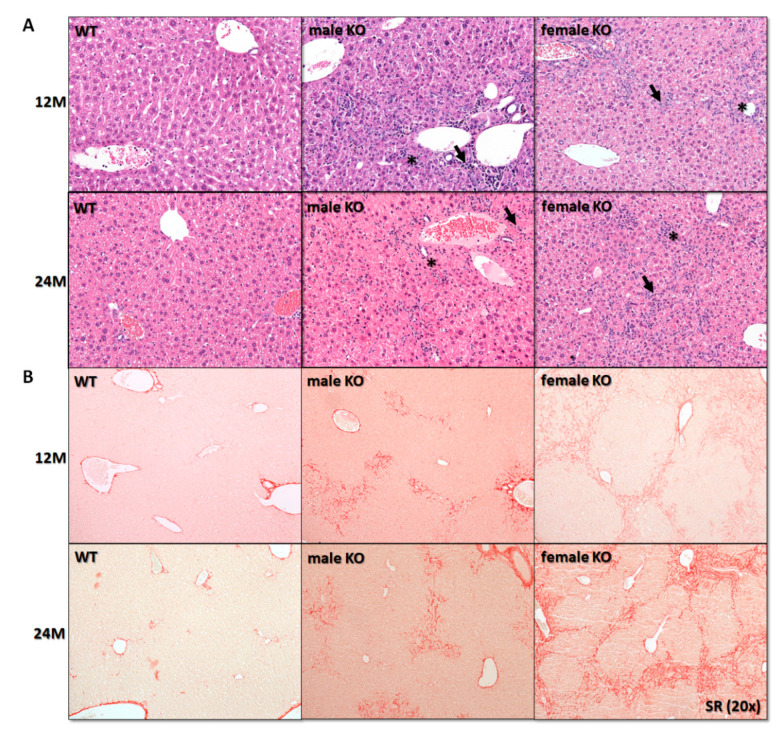
Blocking cholesterol synthesis due to depletion of *Cyp51* resulted in histological alterations of the liver, most prominent in females. (**A**) Histological examination revealed ductular reaction (*) and inflammation (→) (H&E, original magnification ×200), N = 5–10. (**B**) Sirius red stain (SR, original magnification ×200) showed more pronounced fibrosis in female transgenic mice, N = 3–5. M, month; KO, *Cyp51* KO; WT, wild-type.

**Figure 2 cancers-12-03302-f002:**
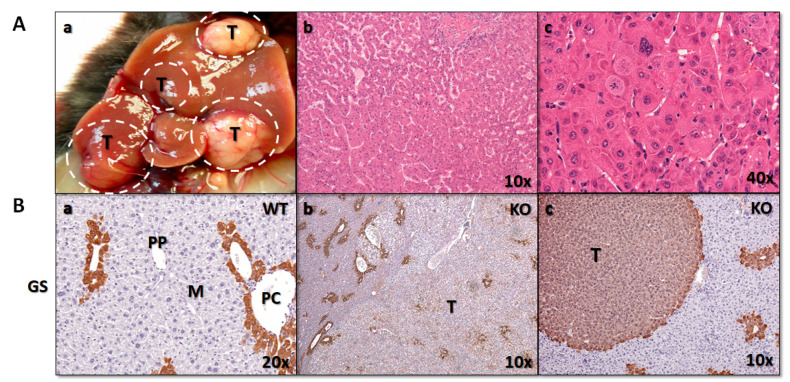
(**A**) Macroscopic and histological presentation of liver tumors. (**a**) Representative gross image of liver tumors in 24M KO mice. (**b**) Representative histologic image of HCC showing architectural abnormalities. (**c**) Higher magnification of (**c**), showing cellular and nuclear polymorphism with brisk mitotic activity. Hematoxylin and Eosin (H&E), original magnification A, B ×100, C ×400. (**B**) Immunohistochemical expression of GS marker of HCC in livers of WTs and KOs. Original magnification, (**a**) ×200, (**b**) ×100, (**c**) ×400. GS, glutamine synthetase; T, tumor; PP, periportal; PC, pericentral, M, midzonal; WT, wild-type; KO, Cyp51 KO; T, tumor.

**Figure 3 cancers-12-03302-f003:**
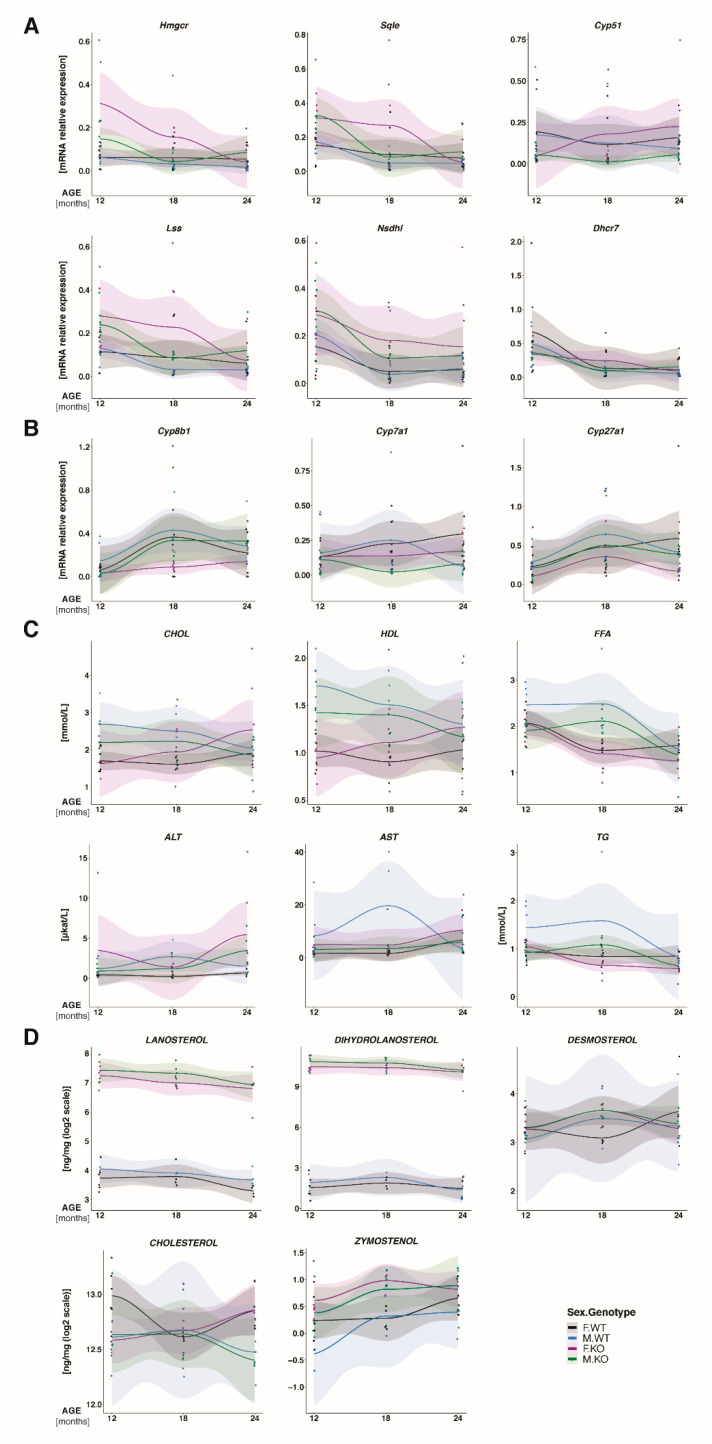
Adaptation to hepatocyte block in cholesterol synthesis during aging. Measurements were performed in KO and WT mice at 12M, 18M and 24M of age. Expression of hepatic genes from (**A**) from cholesterol synthesis (**B**) and bile acid synthesis. N = 5 mice per sex/genotype/age group. (**C**) Plasma concentration of biochemical parameters, N = 5–10 mice per sex/genotype/age group. CHOL—cholesterol, HDL—high-density lipoproteins, FFA—free fatty acids, ALT—alanine aminotransferase, AST—aspartate aminotransferase. (**D**) Liver concentration of sterol intermediates and cholesterol, N = 4–6 mice per sex/genotype/age group. Light color bands—95% confidence interval. Dots—individual measurements. KO, *Cyp51* KO; WT, *Cyp51* WT, F = female; M = male.

**Figure 4 cancers-12-03302-f004:**
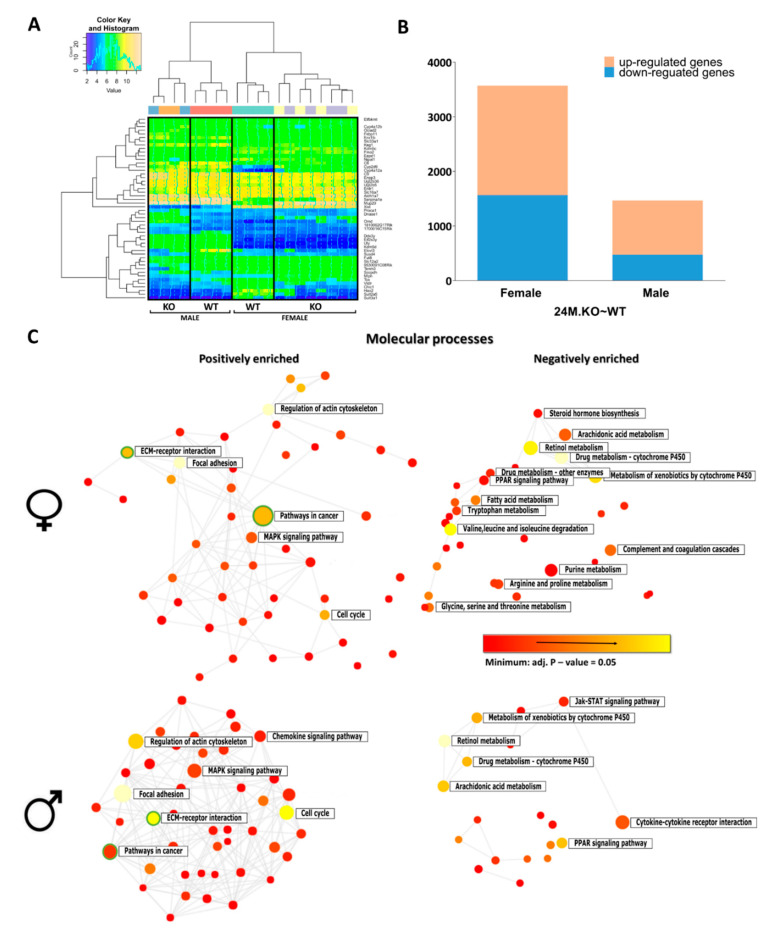
Target genes associated with blocked cholesterol synthesis. (**A**) Heat-map of top 50 DE transcripts that showed significant differences in at least one comparison: hierarchical clustering distinguished between the genotypes and sex of 24M KO (24M.KO~WT) mouse livers. (**B**) Stacked column plots with numbers of differentially up- and down-regulated sex-specific genes of 24M *Cyp51* KO mice. (**C**) The functional enriched networks of up- and down-regulated DE genes in female and male 24M KO mice using NetworkAnalyst. The sizes of the nodes correspond to the number of DE genes within a particular molecular process. All pathway nodules are differentially expressed (*p* < 0.05), the gradation rising from red to yellow (most enriched) in the network of positively (left) and negatively (right) enriched molecular processes. Circles representing cancer pathways and ECM-receptor interaction are surrounded by green lines.

**Figure 5 cancers-12-03302-f005:**
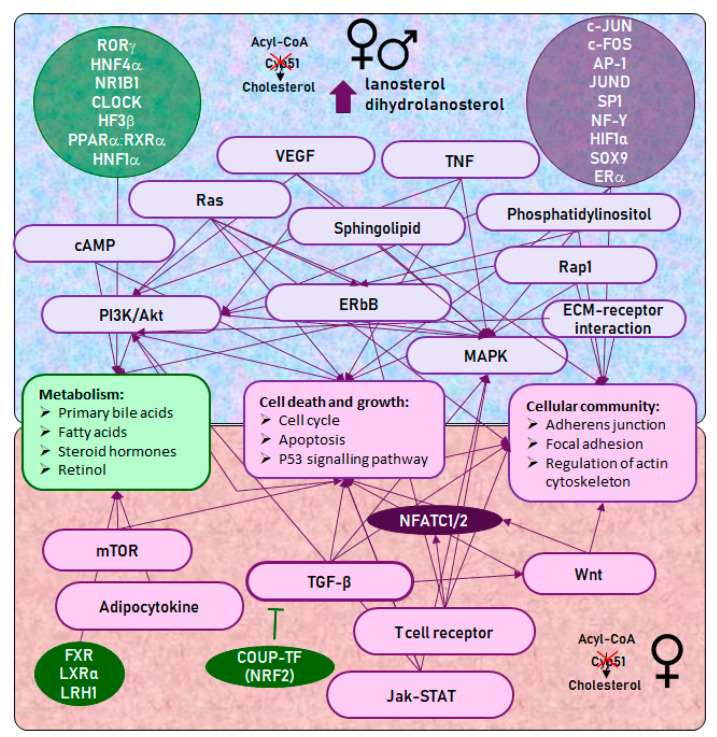
Hepatocyte-specific *Cyp51* KO resulted in the deregulation of multiple signalling pathways and TFs, leading to the development of female-prevalent liver cancer. Arrows indicate connections between KEGG signalling pathways, TFs, and cellular processes. KEGG pathways and TFs are violet if positively enriched and green if negatively enriched at 24M. Top half—selected KEGG signalling pathways and TFs that were deregulated in females and males, leading to the inhibition of basic metabolism and an increase in processes of cell death and growth, and cellular communication. MAPK and PI3K/Akt signalling pathways have a central role and are regulated by other signalling pathways and TFs, all directly or indirectly contributing to carcinogenesis. Bottom half—female-specific KEGG signalling pathways and TFs. TF NFATC1/2 is central to the female-specific phenotype and is connected by the TGF-β, mTOR and Wnt signalling pathways, positively enriched only in female *Cyp51* KO livers.

**Figure 6 cancers-12-03302-f006:**
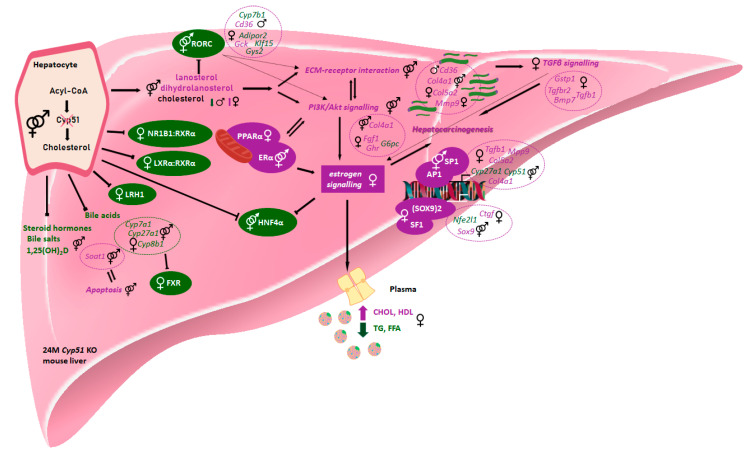
Sex-dependent metabolic and transcriptional changes after disrupted *Cyp51* from cholesterol synthesis align with hepatocarcinogenesis in humans. Arrows indicate active connections between enriched genes, KEGG pathways, and TFs. Blocked arrows represent repression. Hatched arrows represent possible transcription regulation of the selected pathway or connection between selected pathways in KO mice. Metabolites, DE genes, KEGG pathways, and TFs are violet if positively enriched and green if negatively enriched. Icons were used from the Reactome icon library. All selected mouse signalling and metabolic pathways and TFs were aligned with human literature data.

**Table 1 cancers-12-03302-t001:** The most significantly enriched KEGG pathways in 19W KO (19W.KO~WT) and 24M KO (24M.KO~WT) mice. Parameters shown in bold are statistically significant. KEGG, Kyoto Encyclopaedia of Genes and Genomes; * *p* < 0.05, ** *p* < 0.01, *** *p* < 0.001.

KEGG Pathway	19W.KO~WT				24M.KO~WT			
	Females		Males		Females		Males	
	LogFC	adj. *p*-Value	LogFC	adj. *p*-Value	LogFC	adj. *p*-Value	LogFC	adj. *p*-Value
Cell cycle	1.13	**0.01 ***	1.06	0.024	1.60	**<0.001 *****	2.13	**<0.001 *****
Pathways in cancer	1.09	**0.001 ****	0.79	**0.019 ***	1.89	**<0.001 *****	1.84	**<0.001 *****
ECM-receptor interaction	1.30	**0.005 ****	1.50	**0.005 ****	1.82	**<0.001 *****	1.85	**<0.001 *****
Proteoglycans in cancer	1.10	**0.001 ****	0.93	**0.005 ****	1.63	**<0.001 *****	1.61	**<0.001 *****
Apoptosis	1.16	**0.001 ****	0.93	**0.003 ****	1.65	**<0.001 *****	1.49	**<0.001 *****
p53 signalling pathway	0.67	**0.011 ***	0.60	**0.032 ***	1.26	**<0.001 *****	1.33	**<0.001 *****
MicroRNAs in cancer	1.05	**0.001 ****	0.68	**0.026 ***	0.87	**<0.001 *****	0.95	**<0.001 *****
Hepatocellular carcinoma	0.67	**0.001 ****	0.32	0.131	0.95	**<0.001 *****	0.72	**0.006 ****
Circadian entrainment	0.21	0.123	0.13	0.387	0.42	**0.020 ***	0.27	0.197
TGF-β pathway	0.61	**0.002 ****	0.38	0.051	0.45	**0.007 ****	0.14	0.427
Wnt signalling pathway	0.29	0.051	0.10	0.588	0.32	**0.018 ***	0.24	0.118
PI3K/Akt signalling pathway	1.00	**0.006 ****	0.94	**0.016 ***	1.61	**<0.001 *****	1.51	**<0.001 *****
Fatty acid degradation	−1.24	**<0.001 *****	−0.49	0.053	−1.18	**0.001 ****	−1.12	**0.009 ****
Metabolic pathways	−1.63	**0.001 ****	−0.14	0.818	−1.80	**0.009 ****	−0.69	0.367
Peroxisome	−0.69	**0.007 ****	−0.57	**0.033 ***	−1.82	**<0.001 *****	−1.20	**0.018 ***
Primary bile acid biosynthesis	−1.76	**<0.001 *****	−0.60	0.103	−1.29	**0.002 ****	−1.40	**0.004 ****
Biosynthesis of unsaturated fatty acid	−0.47	0.063	−0.05	0.874	−0.87	**0.018 ***	−0.46	0.261

**Table 2 cancers-12-03302-t002:** The most significantly enriched TFs in 19W KO (19W.KO~WT) and 24M KO (24M.KO~WT). Parameters shown in bold are statistically significant. * *p* < 0.05, ** *p* < 0.01, *** *p* < 0.001; n.s.—not significant. SP-1: Specificity protein 1; AP1: Activator protein 1; C/EBPβ: DNA damage-inducible transcript 3; PPARα: Peroxisome proliferator-activated receptor alpha; HIF1α: Hypoxia-inducible factor 1-alpha; NFATC1: Nuclear factor of activated T-cells, cytoplasmic 1; CLOCK: Circadian locomotor output cycles kaput; LXRα: Liver X receptor alpha; RXRα: Retinoid X receptor alpha; HNF4α: Hepatocyte nuclear factor 4 alpha; RORα: RAR-related orphan receptor alpha, RORC: RAR-related orphan receptor gamma; NR1B1: Retinoic acid receptor alpha.

Transcription Factor	19W.KO~WT				24M.KO~WT			
	Females		Males		Females		Males	
	LogFC	adj. *p*-Value	LogFC	adj. *p*-Value	LogFC	adj. *p*-Value	LogFC	adj. *p*-Value
SP1	0.81	**0.002 ****	0.85	**0.001 ****	1.47	**<0.001 *****	1.49	**<0.001 *****
C-JUN	0.93	**0.001 ****	0.75	**0.005 ****	1.42	**<0.001 *****	1.50	**<0.001 *****
JUND	0.92	**0.002 ****	0.72	**0.013 ***	1.05	**<0.001 *****	0.91	**<0.001 *****
C-FOS	1.20	**<0.001 *****	0.98	**<0.001 *****	1.47	**<0.001 *****	1.33	**<0.001 *****
AP1	1.00	**<0.001 *****	0.66	**0.003 ****	1.17	**<0.001 *****	0.88	**<0.001 *****
C/EBPβ	0.52	**0.010 ***	0.76	**<0.001 *****	0.91	**<0.001 *****	0.87	**0.004 ****
PPARα	0.51	**0.04 ***	0.06	0.83	0.60	**0.001 ****	0.22	0.27
SOX9	0.34	0.082	0.30	0.125	0.90	**<0.001 *****	0.63	**0.001 ****
(SOX9)2	0.46	**0.012 ***	0.33	0.057	0.67	**0.005 ****	0.50	0.062
HIF1α	0.33	**0.013 ***	0.23	0.070	0.43	**0.008 ****	0.58	**0.004 ****
NFATC1	0.34	**0.042 ***	0.18	0.284	0.36	**0.035 ***	0.29	0.156
CLOCK	−0.24	0.035	−0.17	0.122	−0.26	**0.031 ***	−0.41	**0.009 ****
LXRα:RXRα	n.s.	n.s.	n.s.	n.s.	−0.56	**0.043 ***	−0.19	0.602
HNF4α	−0.73	**0.007 ****	−0.27	0.290	−1.60	**<0.001 *****	−1.17	**0.019 ***
RORα	−0.72	**<0.001 *****	−0.20	0.256	n.s.	n.s.	n.s.	n.s.
RORC	−0.35	**0.028 ***	−0.11	0.474	−1.68	**<0.001 *****	−2.10	**<0.001 *****
NR1B1	−0.21	0.087	−0.35	**0.007 ****	−0.80	**<0.001 *****	−0.59	**<0.001 *****

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
