# Peer review of "Chronic Disruption of the Late Cholesterol Synthesis Leads to Female-Prevalent Liver Cancer"

_cancers, 2020, doi:10.3390/cancers12113302_

Round 1
Reviewer 1 Report
Coken et al. examined the effect of chronic depletion of cholesterol synthesis on hepatocellular carcinoma (HCC) development using a mouse model with the Cyp51 gene being knockout. Cyp51 knockout could promote HCC tumor development when compared to the control mice at a duration of more than one year in the absence of a cirrhotic background. More importantly, the overall tumor incidence rate showed a gender disparity, in which the female mice are more prone to HCC development upon Cyp51 knockout. To understand the underlying biological processes and mechanisms of this observation, the tumor samples and the adjacent normal tissues were harvested from the experimental animals of both sexes and subjected to transcriptomic analysis for differentially expressed genes as well as correlating with the corresponding levels of sterol metabolite and blood parameters. It was found that knocking out the Cyp51 in the post-squalene metabolism in the cholesterol biosynthesis pathway would cause an accumulation of its substrate including lanosterol and dihyrolanosterol, but not the significant depletion in the Desmosterol, the downstream product. Also, a higher degree of dysregulation of other cholesterol-related genes was found in the hepatocytes of the female mice than the male mice. At the systemic level, an increase in plasma cholesterol and HDL was found in the blood of the female mice. Transcriptomic analysis revealed a positive enrichment of pathways in cancer and ECM interaction and a negative enrichment of metabolic-related processes. TGF-b signaling was suggested to be activated in the female Cyp51 knockout HCC tumor. Transcription factor enrichment analysis revealed significant enrichment of specific sets of transcription factors in Cyp51 knockout tumors of both sexes. The comparative analysis further suggested the repression of specific sets of nuclear receptors in the female knockout HCC tumor.
In general, the overall study design of this manuscript is straightforward. However, the organization and presentation can be further improved for clarity. Although the transcriptomic analysis by comparison between Cyp51 knockout and control tissues is informative, the current study lacks some sort of validation about their proposed model. The authors have tried to cite some relevant findings to support their models, however, as molecular pathways in human cancer are highly context-dependent, validating some of the key findings would strengthen the current study and avoid the study becoming too descriptive. It is especially true when we are trying to pinpoint an important biological question in such a specific biological context: the effect of chronic cholesterol depletion in the liver, the most important organ for cholesterol metabolism, in the female background.
A few comments are listed below:
1.The visualization of Figure 3 should be improved to better contrast the difference between each group, which in its current form is a bit difficult to follow. Also, the changes in the cholesterol pathway gene expression should be further elaborated in more detail instead of just simply mentioning the degree of their changes were more pronounced in the female mice.
2.Table S6 and S9. The authors have listed out some differentially expressed genes specifically dysregulated in the female and male tumors respectively. In addition to the global NetworkAnalyst analysis, it would be worthy for the authors to also manually examine and highlight the associated biological functions of the top-ranking dysregulated genes in terms of significant level as well as absolute fold change to see if this can also derive a more precise insight on the identified differentially expressed genes.
3.Table 1, right: The transcription factor enrichment analysis. It is understandable why the authors analyzed the transcription factor enrichment in CYP51 tumors in both sexes. However, as the study is focusing on the underlying molecular dysregulation of CYP51 tumors in female mice, it would be better to compare, identify and focus on transcription factors that are specifically dysregulated in females but not male mice including (SOX9)2, NFATC1, and LXRa:RXRa.
4.Following the above question, would the authors consider to move Table S10 and present it in parallel with Table 1? As in 24M time point, the overall difference between the transcription factor enrichment between the female and male tumor are pretty limited. Also, the enrichment of RORC in female tumors mentioning in the text is only presented in Table S10. The author should further digest and reorganize their data as a whole to make sure that the key findings could be effectively delivered to the readers.
5.The proposed linkage between the negative enrichment RORC and overexpression of the Cd36 downstream is a bit weak and preliminary. The authors have mentioned that almost all the RORC target genes are downregulated except Cd36. Based on the expression data shown in Table S12, it remains inconclusive that Cd36 could be a novel female-related regulator in hepatocarcinogenesis. The same also applies to SOX9.
Minor points:
The manuscript could be improved by further proofreading for accuracy.
Typo:
Figure legend 5: Line 8, Should be “carcinogenesis” but not cancerogenesis
Author Response
Dear Reviewer 1,
thank you for your comments regarding our manuscript ID: cancers-949182 entitled ˝Chronic disruption. of the late cholesterol synthesis leads to female-prevalent liver cancer˝ written by Kaja Blagotinšek Cokan, Žiga Urlep, Gregor Lorbek, Madlen Matz-Soja, Cene Skubic, Martina Perše, Jera Jeruc, Peter Juvan, Tadeja Režen and Damjana Rozman from 14th September 2020.
We appreciate all your valuable insights and the opportunity to address each of these issues and resubmit the manuscript. Response to all comments and suggestions please find in attachment.
We were requested to provide major revisions, and we heavily revised the manuscript. Therefore, we highlighted all the changes in the document in yellow and the "track changes" function, so the changes are more visible to the reviewers.
We sincerely hope that we have adequately addressed the reviewers' comments and that the revised manuscript will be suitable for publication in your journal. Please, see the attachment.
Yours sincerely,
Damjana Rozman

Reviewer 2 Report
Hepatocellular carcinoma is the most common liver tumor and is a leading cause of cancer-related death worldwide. The authors analyzed the Cyp51 hepatocyte-specific knock out mice. The female aged mice over 18 months are more likely to develop liver tumors than male Cyp51 cKO mice. These are difficult experiments that require long term maintenance of the mouse colony. There are lots of experimental details provided. The text is well written. The data are interesting. The authors measured sterol levels, analyzed the histology of tissue, and performed gene expression analyses.
Minor comments:
- If the mouse has deletion of Cyp51 in liver, how the authors explain differential expression of Cyp 51 in the liver in the Figure 3A.
- Figure 3 shows huge difference between cKO and WT liver in the level of lanosterol and dihydrolanosterol (about 2 fold). The changes in desmosterol, and cholesterol are very small and do not look significant. How does post-lanosterol pathway function? It seems that desmosterol and cholesterol are still produced in similar amounts.
Author Response
Dear Reviewer 2,
thank you for your comments regarding our manuscript ID: cancers-949182 entitled ˝Chronic disruption. of the late cholesterol synthesis leads to female-prevalent liver cancer˝ written by Kaja Blagotinšek Cokan, Žiga Urlep, Gregor Lorbek, Madlen Matz-Soja, Cene Skubic, Martina Perše, Jera Jeruc, Peter Juvan, Tadeja Režen and Damjana Rozman from 14th September 2020.
We appreciate all your valuable insights and the opportunity to address each of these issues and resubmit the manuscript. Response to all comments and suggestions please find in attachment.
We were requested to provide major revisions, and we heavily revised the manuscript. Therefore, we highlighted all the changes in the document in yellow and the "track changes" function, so the changes are more visible to the reviewers.
We sincerely hope that we have adequately addressed the reviewers' comments and that the revised manuscript will be suitable for publication in your journal. Please see the attachment.
Yours sincerely,
Damjana Rozman

Reviewer 3 Report
In this manuscript, the authors investigate mechanisms of female-prevalent hepatocarcinogenesis in Cyp51 knockout mice. They found that tumor development in Cyp51 knockout mice was 2:1 prevalence in females. They also found that the enhanced hepatocarcinogenesis in Cyp51 knockout females was attributed by the sex-dependent metabolic reprogramming of cholesterol-related pathways in aging females. Overall, the manuscript is well written. The concept of this manuscript is good. The finding that Cyp51 deletion alters cholesterol metabolic pathways in a sex-dependent manner and triggers female-prevalent hepatocarcinogenesis could be valuable addition to the field. The reviewer has some minor comments.
- In Fig 2, IHC staining of several markers to distinguish hepatocellular carcinoma (HCC) and cholangiocarcinoma (CC) should be considered, such as HCC marker a-fetoprotein (AFP) and CC markers cytokeratin CK7 or CK19.
- In Fig 3C, Cyp51 knockout results in elevation of hepatic lanosterol and dihydrolanosterol in both males and females. No changes between the sexes were observed in the hepatic accumulation of lanosterol and dihydrolanosterol. The statement on line 340-342 “Our sterol measurements show significant accumulation of CYP51A1 340 substrates lanosterol and dihydrolanosterol, with differences between males and females regarding lanosterol” is incorrect.
- Fig 4C has font so small almost invisible.
- In Table 1, the transcriptional activity of RORC was negatively enriched in 24M KO livers. Why RORC target CD36 was upregulated in KO livers? The authors should discuss it.
- Table S10 demonstrates the enriched KEGG pathways and TFs in a pre-cancerous stage, 19W KO liver. This piece of data should be included in the main manuscript.
Author Response
Dear Reviewer 3,
thank you for your comments regarding our manuscript ID: cancers-949182 entitled ˝Chronic disruption. of the late cholesterol synthesis leads to female-prevalent liver cancer˝ written by Kaja Blagotinšek Cokan, Žiga Urlep, Gregor Lorbek, Madlen Matz-Soja, Cene Skubic, Martina Perše, Jera Jeruc, Peter Juvan, Tadeja Režen and Damjana Rozman from 14th September 2020.
We appreciate all your valuable insights and the opportunity to address each of these issues and resubmit the manuscript. Response to all comments and suggestions please find in attachment.
We were requested to provide major revisions, and we heavily revised the manuscript. Therefore, we highlighted all the changes in the document in yellow and the "track changes" function, so the changes are more visible to the reviewers.
We sincerely hope that we have adequately addressed the reviewers' comments and that the revised manuscript will be suitable for publication in your journal. Please see the attachment.
Yours sincerely,
Damjana Rozman

Round 2
Reviewer 1 Report
The overall quality of the manuscript has been improved after revision. The authors have to make sure that the abstract appeared in the submission system is the same as the one included in the revised manuscript. Please also further confirm whether the Cd36 overexpression is a new female- (stated in the previous version) or male-related (stated in the revised version) regulator of ECM-receptor signalling in hepatocarcinogenesis before finalizing the manuscript.
